

# Characterization, development and multiplexing of microsatellite markers in three commercially exploited reef fish and their application for stock identification

Laura Taillebois[1,2], Christine Dudgeon[3], Safia Maher[3], David A. Crook[1], Thor M. Saunders[4], Diane P. Barton[1,4], Jonathan A. Taylor[4], David J. Welch[5], Stephen J. Newman[6], Michael J. Travers[6], Richard J. Saunders[7,8] and Jennifer Ovenden[3]

[1] Research Institute for the Environment and Livelihoods, Charles Darwin University, Darwin, Northern Territory, Australia
[2] North Australia Marine Research Alliance, Arafura Timor Research Facility, Darwin, Northern Territory, Australia
[3] Molecular Fisheries Laboratory, School of Biomedical Sciences, University of Queensland, Brisbane, Queensland, Australia
[4] Fisheries Research, Northern Territory Department of Primary Industries and Fisheries, Berrimah, Northern Territory, Australia
[5] C$_2$O Fisheries, Cairns, Queensland, Australia
[6] Western Australian Fisheries and Marine Research Laboratories, Department of Fisheries, Government of Western Australia, North Beach, Western Australia, Australia
[7] Centre for Sustainable Tropical Fisheries and Aquaculture, James Cook University, Douglas, Queensland, Australia
[8] Animal Science, Queensland Department of Agriculture and Fisheries, Brisbane, Queensland, Australia

Corresponding author
Laura Taillebois,
laura.taillebois@cdu.edu.au

## ABSTRACT

Thirty-four microsatellite loci were isolated from three reef fish species; golden snapper *Lutjanus johnii*, blackspotted croaker *Protonibea diacanthus* and grass emperor *Lethrinus laticaudis* using a next generation sequencing approach. Both IonTorrent single reads and Illumina MiSeq paired-end reads were used, with the latter demonstrating a higher quality of reads than the IonTorrent. From the 1–1.5 million raw reads per species, we successfully obtained 10–13 polymorphic loci for each species, which satisfied stringent design criteria. We developed multiplex panels for the amplification of the golden snapper and the blackspotted croaker loci, as well as post-amplification pooling panels for the grass emperor loci. The microsatellites characterized in this work were tested across three locations of northern Australia. The microsatellites we developed can detect population differentiation across northern Australia and may be used for genetic structure studies and stock identification.

## INTRODUCTION

Microsatellites are hypervariable, nuclear-encoded and codominant-inherited markers used for a variety of aquaculture and fisheries applications, including determining the spatial extent of fisheries stocks and other important applications of population genetics. *De novo* discovery of microsatellites is required for analyses in the laboratory with each non-model species studied; however the costs are high and the procedure involving cloning is time-consuming (*Peters et al., 2009*). The alternative to *de novo* development is cross-species amplification where existing microsatellite loci of related species are used on the target species; but this is often hampered by the lack of conserved flanking sequences of microsatellites or the lack of data on related species. The adoption of Next-Generation Sequencing (NGS) by researchers using microsatellite loci has made the discovery of microsatellite markers easier (*Gardner et al., 2011*) and is becoming the preferred method for developing microsatellites (*Abdelkrim et al., 2009*; *Castoe et al., 2010*; *Fernandez-Silva et al., 2013*). Once the microsatellites are identified, major cost and time reductions in the laboratory are achieved through polymerase chain reaction (PCR) multiplexing. The challenge of PCR multiplexing is to combine several microsatellite primers into one PCR cocktail to amplify several microsatellite loci at the same time.

Herein, we describe the discovery, characterization, development and multiplexing of microsatellite loci of three reef fish species of commercial and recreational significance: golden snapper (*Lutjanus johnii*, Lutjanidae), blackspotted croaker (*Protonibea diacanthus*, Sciaenidae) and grass emperor (*Lethrinus laticaudis*, Lethrinidae). *Lutjanus johnii,* is a highly prized sport and food fish and is harvested in the commercial, recreational, charter and indigenous sectors of northern Australia and many other fisheries worldwide (*Allen, Swainston & Ruse, 1997*). The catch of *L. johnii* has been declining in the Northern Territory since 1997 and this species is considered overfished (*Grubert et al., 2013*; *Saunders et al., 2014a*). Sciaenids form the basis of commercial and recreational fisheries of tropical and temperate regions worldwide (*Lenanton & Potter, 1987*; *Rutherford et al., 1989*) and several large species are considered threatened or vulnerable due to over-fishing (*Rao et al., 1992*; *Saunders et al., 2014b*; *True, Loera & Castro, 1997*). Among Sciaenid species, *P. diacanthus* is vulnerable to over-exploitation because of its predictable aggregating behavior (*Bowtell, 1995*; *Bowtell, 1998*; *Phelan, Gribble & Garrett, 2008*). *Lethrinus laticaudis* is considered an excellent eating fish and is targeted by commercial fishers and recreational anglers across northern Australia (*Coleman, 2004*). Although *L. laticaudis* is considered robust to fishing pressure (*Grubert, Kuhl & Penn, 2010*) due to its high reproductive capacity (i.e., serial batch spawners, high spawning frequency, high batch fecundity) (*Ayvazian et al., 2004*), it is heavily exploited in some areas. These three fish species are of high economic value and the sustainability of the fisheries they support is potentially threatened by over harvesting and thus requires the development of suitable management programs. The development of genetic tools is necessary to further investigate their population genetics and assess stock structure.

In this study, we provide novel polymorphic microsatellite loci for the three species. We also describe a fast and cost-effective protocol for species-specific microsatellite marker

discovery using genomic sequencing and multiplexing. Finally, we explore the relevance of the described microsatellite markers for further population genetics by looking at the genetic differences found between two locations in the Northern Territory for the three study species. This will inform us on the potential to use these markers for the identification of stocks for management purposes. This is the first report of the nuclear genomes of the three study species and provides useful baseline information for future genetic studies of these important species.

## MATERIALS AND METHODS

### Sample and extraction

Samples selected for the production of the microsatellite loci were derived from muscle tissue collected by the Northern Territory Department of Primary Industries and Fisheries and the Western Australian Department of Fisheries under Charles Darwin University Animal Ethics permit A13014. The *L. johnii* sample was a 210 mm male caught at 6 m depth in Darwin Harbour, Northern Territory, Australia (Middle Arm, 130°58′0.24″E, 12°39′0.97″S) in 2013. The *P. diacanthus* sample was a 890 mm male caught in Fenton patches, Northern Territory, Australia (130°42.084′E, 12°10.664′S) in 2013. The *L. laticaudis* sample (WAM16-001) was a 419 mm male collected from East of the Lacepede Islands, Western Australia, Australia in 2013. Genomic DNA from *L. johnii* and *P. diacanthus* was extracted using Qiagen DNeasy Blood & Tissue columns (Qiagen, Germantown, MD, USA) following the manufacturer's instructions. *Lethrinus laticaudis* genomic DNA was extracted using a salting-out method as described in *Broderick et al. (2011)*. Genomic DNA from all samples for testing the loci and further genotyping was extracted using ISOLATE II Genomic DNA Kit (Bioline) following the manufacturer's instructions. This resulted in 100 µL of eluted DNA for each sample. All the DNA extracts were quantified using the Qubit v3 (ThermoFisher) fluorometric method.

### Next-generation sequencing and primer selection

The purified genomic DNA of *L. johnii* and *P. diacanthus* was prepared for direct shotgun sequencing using the Iron ExpressTM fragment library kit and sequenced on an IonTorrent Personal Genome Machine using an Ion318 chip (Life Technologies Corporation, Grand Island, NY, USA). The purified genomic DNA of *L. laticaudis* was sequenced on an Illumina® MiSeq as part of a 2 × 300 bp run at the Australian Genome Research Facility. Because two different NGS platforms were used to scan the genomes of the three species we were able to compare their performance for microsatellite design and to assess whether equivalent results were obtainable from each platform. *Lutjanus* and *Lethrinus* genera and Sciaenidae are known to have genome size comparable to other fish species (average size for *Lutjanus* = 1,066 Mb, *Lethrinus* = 1,192 Mb, Sciaenidae = 753 Mb, Perciformes = 919 Mb; *Gregory, 2001*). The paired-end reads obtained with the MiSeq run were merged using FLASH source code (*Magoč & Salzberg, 2011*) and their quality was checked in FastQC (*Andrews, 2010*); the first 10 bp were trimmed in Geneious v 9.0 (*Kearse et al., 2012*).

From the NGS data we looked for sequences longer than 300 bp that contained a microsatellite repeat that would be suitable for primer design. These sequences were checked

for microsatellite motifs and forward and reverse primers were designed using the software QDD2 beta (*Meglécz et al., 2010*). Sequences with target microsatellites and primers were then filtered according to the following criteria: only pure repeats were selected; all dinucleotide repeats were excluded; repeats greater than eight were selected; loci with a predictive target sequence length above 300 bp were selected; primers with a distance less than 20bp from the repeat sequence were excluded; and the PCR primers with a PCR_PRIMER_ALIGNSCORE equal or above 6 were excluded to discard primers with high alignment scores to the amplicon. A unique pair of primers was selected for each locus. The PCR predicted sequences for all the loci were imported into Geneious v 9.0 and blasted (MEGABLAST) against the NCBI GenBank database to check if the microsatellites fell into coding regions. Sequences that would be homologous to any other NCBI sequence likely to be functional were excluded. All the primers were blasted against their original genomic database built using the NGS reads. Only microsatellites with primers that had one hit across the whole genome were kept for further steps to increase the chance of each primer to amplify a unique sequence. As a final check point before wet laboratory work and to make sure each pair of primers bound to the 5′ and 3′ ends of a unique sequence containing a microsatellite we selected, containing microsatellites sequences and pairs of primers were mapped *de novo*. For each species, we selected the 48 microsatellites that contained the best quality repeats with the highest number of tri- tetra- or penta-nucleotide repeats possible and with no small dinucleotide repeats between the primer and the microsatellite sequence to avoid any noise that may interfere with scoring genotypes.

Forward primers were tagged on the 5′ end with the universal CAG sequence (5′-CAGTCGGGCGTCATCA-3′). Inclusion of the 5′-tail will allow use of a CAG-tagged universal primer in the PCR that is fluorescently labeled for detection on the sequencing machine (*Schuelke, 2000*). Additionally, a pig-tail (5′-GTTTCTT-3′) was added to the 5′ end of the reverse primers to increase the accuracy of genotyping and ensure the consistency of the amplicon size (*Brownstein, Carpten & Smith, 1996*). The resulting 48 pairs of primers were synthetized by Integrated DNA Technologies (www.idtdna.com).

## Loci and primers testing

For each species, the 48 pairs of primers were tested over a set of genomic DNA extracted from eight individuals of the target species. The fluorescent dye labelling of PCR fragments in one reaction was performed with three primers. Amplification reactions were carried out in a 8.8 μL volume comprising 1 μL of DNA, 4.84 μL of 2× Bioline Taq Mastermix, 4.4 pmol of locus-specific forward CAG-tagged primer, 22 pmol of locus-specific reverse pig-tail-tagged primer and 22 pmol of universal CAG primer labeled with a 6-FAM fluorescent dye. The heating cycle parameters used for amplification were 95 °C for 3 min, 37 cycles of denaturation at 94 °C for 15 s, annealing for 15 s at 57 °C and elongation at 72 °C for 60 s. A final extension at 72 °C for 30 s was performed. Post-amplification, the PCR products were diluted with water 1:20. We added 2 μL of these diluted PCR products to 10 μL of Hi-Di formamide (ABI) and 0.05 μL of GenScan-500 LIZ (ABI) size standard. Samples were denatured at 95 °C for 3 min and sized on the ABI 3730xl capillary sequencer (Applied Biosystems, Carlsbad, California, USA) using the conditions

set down by the manufacturer. Chromatograms were analysed using Geneious v 9.0 (*Kearse et al., 2012*).

Criteria used to select the best loci among the 48 tested for each species included the amplification success rate, peak intensity, the presence or absent of stutter peaks, the polymorphism of the loci, the number of alleles and heterozygosity. The best loci were individually tested against a further 23 samples of the target species.

## Multiplex optimization of PCR

In order to reduce the cost and time of genotyping for further genetic studies, the newly designed microsatellites were combined into multiplex panels of 2–4 loci. The panels were set up based on the microsatellite allele-size range. The primers for all the loci of each panel were combined in a single PCR to allow the amplification of several microsatellite loci at the same time. When allele-size ranges overlapped, alternate dyes were employed to allow the discrimination of each locus on the chromatograms. Each of the four ABI dyes FAM, VIC, NED and PET were tailed with a unique M13 primer: M13F (5′-TTTCCCAGTCACGACGTTG-3′), M13V (5′-GCGGATAACAATTTCACACAGG-3′), M13N (5′-TAAAACGACGGCCAGTGC-3′) and M13P (5′-CACAGGAAACAGCTATGACC-3′). The 5′ end of the forward primer for the locus was synthetized with the corresponding M13 tail to allow fluorescent labeling of PCR product using a 3-primer PCR protocol as described above (*Schuelke, 2000*). Several multiplex trials were conducted to find the best combination of loci with the optimal concentration of primers and PCR parameters. Primer pairs that failed to amplify within a multiplex were removed from the panels and further optimization focused on the remaining primer pairs. For each species, the multiplex trials were all evaluated against eight samples that were the same for those used in the 23 samples above, allowing the consistency to be checked across the templates.

## Genetic variability and population genetics

In order to test if the herein developed microsatellites would be useful to discriminate fish stocks across northern Australia we collected samples from two locations in the Northern Territory and one location in Western Australia and assayed their population structure. For each of the three species, samples were collected from Camden Sound (Western Australia, Australia), Wadeye (Northern Territory, Australia) close to the Western Australia border and Vanderlin Islands in the bottom of the Gulf of Carpentaria (Northern Territory, Australia) (Fig. 1). Fourteen *L. johnii*, eighteen *P. diacanthus* and twenty-eight *L. laticaudis* were collected from Camden Sound (Western Australia, Australia); twenty-nine *L. johnii*, twenty-five *P. diacanthus* and twenty-seven *L. laticaudis* were collected from Wadeye (Northern Territory, Australia); twenty-five *L. johnii*, twenty-nine *P. diacanthus* and twenty-nine *L. laticaudis* were collected from Vanderlin Islands (Northern Territory, Australia). All individuals were assayed as part of the multiplex panels or PCR pooled. The multiplex PCR were comprised of 2–5 μL of DNA depending on the samples (approximately 20 ng total), 8 μL of 2× Bioline Taq Mastermix and 4 μL of panel primer mix containing various quantities of primers as described in Table 1. Concentrations of the different
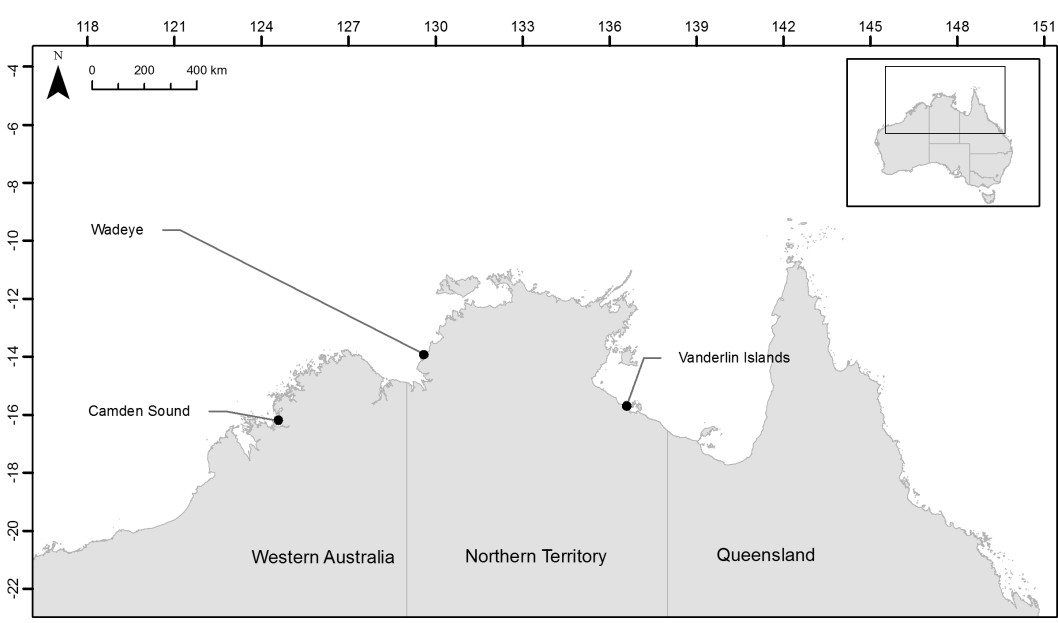

**Figure 1   Location of the three sampling sites across northern Australia.**

primers were adjusted to obtain homogenous PCR products revealed by similar intensity chromatogram peaks for each of the dyes within each panel. The heating cycle parameters, sizing of the alleles and chromatogram analyses were conducted using the same method as stated above.

The obtained datasets were statistically evaluated. The potential for null alleles, large allele dropout and stuttering to interfere with scoring accuracy was evaluated for each microsatellite locus in each sample using Microchecker v.2.2.3 (*Van Oosterhout et al., 2004*). The software Arlequin 3.5.2.2 (*Excoffier & Lischer, 2010*) was used to calculate the number of alleles ($A$), expected ($H_e$) and observed ($H_o$) heterozygosity and conduct exact tests of conformance of genotypic proportions to Hardy–Weinberg equilibrium expectations. Estimation of probability values ($P_{HW}$) employed a Monte Carlo Markov Chain (MCMC) of $10^5$ steps and $5 \times 10^4$ dememorization. Genotypic equilibrium between pairs of microsatellites (linkage disequilibrium) was tested in Arlequin with 10,000 permutations. Fixation indices ($F_{ST}$) between pairs of sample localities were estimated as implemented in Arlequin to identify possible spatial boundaries among sample locations.

## RESULTS AND DISCUSSION

The IonTorrent sequencing technology allowed the production of longer reads (range 8–620 bp) compared to the Illumina MiSeq that produced 300 bp reads fixed by the method. The paired-end reads of the MiSeq run were merged to increase their length to 300–575 bp and allow the detection of at least 300 bp length sequences that may contain a microsatellite locus. This resulted in 1,169,198 reads, which is less than what was used to select microsatellite loci in the first two species. The MiSeq run produced higher quality sequences than the IonTorrent (Phred score: 36 vs. 29). Quality profiles along the

**Table 1  Technical details on the multiplex polymerase chain reaction (PCR) and post-PCR pooled products of microsatellite loci in *Lutjanus johnii*, *Protonibea diacanthus* and *Lethrinus laticaudis*.** Included in the table are the multiplex PCR panels for *L. johnii*, *P. diacanthus* and post-PCR pooled products panels for *L. laticaudis*, primer mix quantities per reaction (μL) within each multiplex and fluorescent dye labels used for each locus in the PCR reactions.

| Species | Panel | Microsatellite | Quantity (μL) | ABI dye |
|---|---|---|---|---|
| *Lutjanus johnii* | 1 | *Luj094* | 0.2 | FAM |
| | | *Luj068* | 0.6 | PET |
| | | *Luj027* | 0.6 | FAM |
| | | *Luj076* | 0.6 | VIC |
| | 2 | *Luj051* | 0.2 | VIC |
| | | *Luj090* | 0.4 | FAM |
| | | *Luj114* | 0.6 | VIC |
| | | *Luj091* | 0.6 | PET |
| | 3 | *Luj072* | 0.7 | FAM |
| | | *Luj082* | 0.7 | VIC |
| *Protonibea diacanthus* | 1 | *Prd012* | 0.2 | PET |
| | | *Prd023* | 0.2 | VIC |
| | | *Prd044* | 0.2 | FAM |
| | | *Prd042* | 0.2 | NED |
| | 2 | *Prd018* | 0.4 | NED |
| | | *Prd045* | 0.2 | PET |
| | | *Prd046* | 0.2 | FAM |
| | | *Prd020* | 0.2 | PET |
| | 3 | *Prd036* | 0.2 | VIC |
| | | *Prd049* | 0.2 | FAM |
| | | *Prd024* | 0.2 | PET |
| *Lethrinus laticaudis* | 1 | *Lel033* | – | NED |
| | | *Lel012* | – | PET |
| | | *Lel040* | – | VIC |
| | | *Lel011* | – | FAM |
| | 2 | *Lel032* | – | NED |
| | | *Lel028* | – | PET |
| | | *Lel041* | – | VIC |
| | | *Lel013* | – | FAM |
| | 3 | *Lel036* | – | NED |
| | | *Lel039* | – | PET |
| | | *Lel044* | – | VIC |
| | | *Lel047* | – | FAM |
| | 4 | *Lel027* | – | FAM |

**Notes.**

Primer mix for each locus is all initially made with 6 μL of forward (M13) primer (10 μM), 30 μL of reverse primer (10 μM), 30 μL of M13 labeled dye (10 μM) and 84 μL of water. The primer mixes of each panel are mixed in the proportion given in the table with water added to reach 4 μL.

IonTorrent reads showed that the quality of the sequencing decreased with length meaning that the end of the longer reads (>325 bp) had a lower quality then at their start. QDD Pipe1 detected between 110,000 and 170,000 sequences containing a microsatellite sequence depending on the species (Table 2). This number was independent of the type of NGS platform used. From those sequences QDD Pipe2 removed the low complexity sequences (no BLAST to itself), putative minisatellites (short sequences of repeated nucleotides) and sequences that had BLAST hit to other sequences to only keep the singletons and unique consensus sequences. QDD Pipe3 designed primers for all QDD Pipe2 output reads. The resulting number of sequences that contained a microsatellite sequence and the corresponding primers were given in the final output of QDD pipeline, and varied between 20,000 and 30,000 depending on the species (Table 2). After applying the filtering criteria described previously, 97 potentially amplifiable microsatellite reads were found for *P. diacanthus*, 121 for *L. johnii* and 103 for *L. laticaudis*. From those microsatellite reads, we selected the ones with the smallest number of repeats but greater than eight and eliminated those with small repeats between the primer and the microsatellite to reach 48 microsatellite loci per species being ultimately tested in the laboratory.

The testing of 144 primer pairs resulted in the selection of 34 polymorphic loci that could be reliably scored and showed consistent amplification success. We selected a final set of 10 loci for *L. johnii*, 11 loci for *P. diacanthus* and 13 loci for *L. laticaudis* (Table 3). Multiplex panels of microsatellites were developed for the two species *L. johnii* and *P. diacanthus*, and the optimization of each panel resulted in the efficient assay and unambiguous scoring of microsatellites in the two species. Although the M13 labeling system worked very well for *L. johnii* and *P. diacanthus* it did not amplify successfully as part of PCR multiplexes for *L. laticaudis*. The reasons why it did not work well for this species are still unclear as the quality of the DNA was even across the three species and the same protocol was followed. However, in-house experiments showed that lengthening the labeled forward primer might facilitate the PCR reaction when multiplexing several loci. Direct labeling of the forward primer may also be another option for multiplexing a large number of loci. For *L. laticaudis*, the loci were all amplified in individual PCR with the CAG labeling system as described above. The resulting PCR products were then pooled according to the panels described in Table 1 before the ABI run.

Genotypes from 10 microsatellites were obtained from multiplexed PCR for 68 individuals of *L. johnii*. There was 1.91 % missing data and the number of alleles for each locus varied between 4 and 23 (Table 3). Microchecker indicated the possible occurrence of null alleles at location Wa for locus *Luj012* and at CS for locus *Luj018* with possible stuttering or scoring errors for the latter. There were only 3/45 significant tests for linkage disequilibrium between pairs of loci (*Luj094* × *Luj051*, *Luj051* × *Luj091* and *Luj076* × *Luj082*) and there was no deviation from Hardy–Weinberg equilibrium (HWE) detected. Heterozygosity was moderate to high for all loci (mean overall loci 0.707 ± 0.200) and generally similar to expectations (around 0.7 for marine fish, *DeWoody & Avise, 2000*). Genotypes from 11 microsatellites were obtained from multiplexed PCR for 72 individuals of *P. diacanthus*. There was 2.77 % missing data and the number of alleles for each locus varied between five and 19 (Table 3). Microchecker indicated the possible occurrence of

**Table 2  Next-generation sequencing and bioinformatics details obtained from FastQC software (*Andrews, 2010*) and QDD pipeline (*Meglécz et al., 2010*) for *Lutjanus johnii*, *Protonibea diacanthus* and *Lethrinus laticaudis*.**

|  | *Lutjanus johnii* | *Protonibea diacanthus* | *Lethrinus laticaudis* |
|---|---|---|---|
| Genomic DNA extraction | Qiagen DNeasy | Qiagen DNeasy | Salting out |
| NGS Technology | IonTorrent \| Ion318Chip | IonTorrent \| Ion318Chip | Illumina® MiSeq |
| Library preparation | Iron ExpressTM | Iron ExpressTM |  |
| Type of reads | Single reads | Single reads | Paired-end reads |
| Number of reads | 1,374,891 | 1,587,789 | 2,800,640 |
| Merged reads | – | – | 1,169,198 |
| Reads length | 8–620 | 8–618 | 300 |
| Merged reads length | – | – | 300–575 |
| **FASTQC** |  |  |  |
| % GC | 41 | 42 | 39 |
| Sequence quality < Phred 20 | yes at positions >325 bp | yes at positions >350 bp | no |
| Per sequence quality—Phred score | 29 | 29 | 36 |
| Sequence length distribution | peak at 350 bp | peak at 350 bp | plateau at 520–540 bp |
| **QDD2** |  |  |  |
| **QDD2 pipe 1—Sequence preparation and microsatellite detection** |  |  |  |
| Number sequence length ≥80 bp | 1,235,685 | 1,405,082 | 1,169,198 |
| Number sequence with microsatellite | 109,641 (8.9%) | 167,702 (11.9%) | 130,269 (11.1%) |
| **QDD2 pipe 2—Sequence similarity detection** |  |  |  |
| Total # input sequences | 109,641 | 167,702 | 130,269 |
| Numer of unique consensus sequences | 18,978 | N/A | N/A |
| Number of singleton sequences | 49,122 | N/A | N/A |
| Number of reads in output | 68,100 | 63,789 | 69,714 |
| **QDD2 pipe 3—Primer design** |  |  |  |
| Total number of input sequences | 68,100 | 63,789 | 69,714 |
| Total number of sequences with target MS | 67,461 | 63,785 | 69,714 |
| Total number of sequences with primers | 29,485 | 20,233 | 19,867 |
| **Filtering QDD output** |  |  |  |
| Total # input sequences | 29,485 | 20,233 | 19,867 |
| Total # sequences after filtering criteria | 121 | 97 | 103 |

null alleles at location CS for locus *Prd068* and at VI for locus *Prd012*. There were only 3/55 significant tests for linkage disequilibrium between pairs of loci (*Prd046 × Prd018, Prd020 × Prd018 and Prd018 × Prd045*) and overall deviations from Hardy–Weinberg equilibrium (HWE) were detected at two loci *Prd023* ($p$-value = 0.034) and *Prd018* ($p$-value = 0.008). Heterozygosity was variable and with an overall mean lower than for *L. johnii* (0.673 ± 0.185). Genotypes from 10 microsatellites were obtained from pooled post PCR products for 84 individuals of *L. laticaudis* as three of our developed microsatellites did not amplified consistently in all the samples. There was 2.61% missing data and the number of alleles for each locus varied between 9 and 22 (Table 3). Microchecker indicated the possible occurrence of null alleles at location CS for loci *Lel033* and *Lel012*. There were 6/78 significant tests for linkage disequilibrium between pairs of loci (*Lel040 × Lel012, Lel040 × Lel032, Lel011 × Lel027, Lel012 × Lel032, Lel012 × Lel013 and Lel028 × Lel027*)

Taillebois et al. (2016), *PeerJ*, DOI 10.7717/peerj.2418

**Table 3** Characteristics of the 34 microsatellite markers developed in *Lutjanus johnii*, *Protonibea diacanthus* and *Lethrinus laticaudis*.

| Species | Locus | | Primer sequences (5′–3′) (fluorescent label) | Repeat motif | GenBank accession no. | n | Allele size range (bp) | #A | $H_O$ | $H_E$ | $P_{HW}$ |
|---|---|---|---|---|---|---|---|---|---|---|---|
| *Lutjanus johnii* | Luj027 | F: | CTGGGCCACACTGATAAAGC (FAM) | (AGC)9 | KX387441 | 68 | 152–179 | 8 | 0.309 | 0.306 | 0.606 |
| | | R: | GGCTCTGAACCTGGGAGATT | | | | | | | | |
| | Luj094 | F: | TCTCAGAGGGTTTGATGCAG (FAM) | (AATC)9 | KX387437 | 68 | 223–239 | 4 | 0.426 | 0.470 | 0.205 |
| | | R: | CTTTGGCGCTTTCTATCAGC | | | | | | | | |
| | Luj076 | F: | CGGGTCGAGTCTGTTTGTGT (VIC) | (AAG)15 | KX387436 | 66 | 200–233 | 10 | 0.818 | 0.811 | 0.289 |
| | | R: | CTTCAGACGGATTAGCAGCA | | | | | | | | |
| | Luj068 | F: | CCTAGGGTGTCAGTGTCAGTCA (PET) | (AAAG)20 | KX387435 | 68 | 174–258 | 18 | 0.882 | 0.936 | 0.212 |
| | | R: | TGCCTGTATGTTCTCTTGAGC | | | | | | | | |
| | Luj090 | F: | ATCCTAATGCATCGTGCTTG (FAM) | (AGC)17 | KX387444 | 68 | 194–278 | 23 | 0.868 | 0.922 | 0.146 |
| | | R: | GGCATGTTCTATTGAGGTTGG | | | | | | | | |
| | Luj051 | F: | TGCAGAGCAACAGAACAACAC (VIC) | (ACTG)10 | KX387440 | 67 | 172–192 | 6 | 0.687 | 0.596 | 0.377 |
| | | R: | CACCTTGCGTTTGCAGTCT | | | | | | | | |
| | Luj114 | F: | CCATAACTGCTGTTCTGTATCTGG (VIC) | (AGC)9 | KX387442 | 68 | 276–314 | 11 | 0.794 | 0.739 | 0.725 |
| | | R: | AATACGGCAGATCTCGGGTT | | | | | | | | |
| | Luj091 | F: | TCATTCCCAGGAGCTCAAAT (PET) | (ACAG)12 | KX387438 | 64 | 219–279 | 12 | 0.813 | 0.782 | 0.499 |
| | | R: | AATCGTCACTTTCGACCCAC | | | | | | | | |
| | Luj072 | F: | ACTCGAAGAACACAGCCCAC (FAM) | (AGC)9 | KX387443 | 65 | 192–204 | 5 | 0.738 | 0.675 | 0.052 |
| | | R: | CACATTTGAATCCTTGCTGG | | | | | | | | |
| | Luj082 | F: | AAGTACATCGGAGGGCTGAG (VIC) | (ACGAT)12 | KX387439 | 65 | 220–275 | 12 | 0.800 | 0.833 | 0.548 |
| | | R: | TGTTATCAAAGTTCACCGATACAAA | | | | | | | | |
| *Protonibea diacanthus* | Prd044 | F: | ACAAAGTTTCCTCCTCTGGC (FAM) | (AAG)13 | KX387452 | 71 | 181–211 | 11 | 0.746 | 0.804 | 0.324 |
| | | R: | CACGTTCCATCTTTATTTATTTGC | | | | | | | | |
| | **Prd023** | **F:** | **TCGTGTGAACACTTTGATGC (VIC)** | **(ATC)11** | **KX387448** | **72** | **292–316** | **9** | **0.875** | **0.847** | **0.034** |
| | | R: | CTCGTCTCTGCTCTTGGTCC | | | | | | | | |
| | Prd042 | F: | TACCTTTGAGATGCGAGCG (NED) | (AGC)12 | KX387451 | 72 | 230–248 | 7 | 0.694 | 0.698 | 0.526 |
| | | R: | GTCAAAGCCATCAATCCAGC | | | | | | | | |
| | Prd012 | F: | AGGCTGTTTGAACTGCAGGG (PET) | (AAAG)20 | KX387445 | 64 | 195–271 | 19 | 0.828 | 0.898 | 0.324 |
| | | R: | CATGCTGAGCAATATGTGGG | | | | | | | | |

| Species | Locus | Primer sequences (5′–3′) (fluorescent label) | Repeat motif | GenBank accession no. | n | Allele size range (bp) | #A | $H_O$ | $H_E$ | $P_{HW}$ |
|---|---|---|---|---|---|---|---|---|---|---|
| | Prd046 F: | TCATCCTGAGTTTGTGCTGG (FAM) | (AGC)9 | KX387454 | 72 | 224–236 | 5 | 0.347 | 0.351 | 0.693 |
| | R: | CATGAGTAAGCAGAGCGTGG | | | | | | | | |
| | **Prd018 F:** | **ATGAACGGCATCAGTCAGC (NED)** | **(ACAG)9** | **KX387446** | **68** | **179–207** | **8** | **0.662** | **0.784** | **0.008** |
| | R: | CGTCTGATAAACAGCACTGCC | | | | | | | | |
| | Prd020 F: | CAATGTTCTGCAAGAGCTGC (PET) | (ATC)11 | KX387447 | 71 | 189–216 | 10 | 0.746 | 0.713 | 0.837 |
| | R: | TCAAATGTCAAAGTCCAGTCC | | | | | | | | |
| | Prd045 F: | GTCTATCCATGTTCCAGCCC (PET) | (ATCC)11 | KX387453 | 64 | 279–309 | 8 | 0.672 | 0.639 | 0.400 |
| | R: | TCATCCCAAAGTGACCAACC | | | | | | | | |
| | Prd049 F: | CCTTGTCCTCCTTTCAGGC (FAM) | (ACC)9 | KX387455 | 72 | 216–234 | 5 | 0.278 | 0.316 | 0.280 |
| | R: | GGGTCATTAAACATGGCAGC | | | | | | | | |
| | Prd036 F: | TCACGTGAAGCGTCTACAGC (VIC) | (AAG)12 | KX387450 | 72 | 227–257 | 9 | 0.653 | 0.677 | 0.746 |
| | R: | AAAGGAGGAAACACAGAGCC | | | | | | | | |
| | Prd024 F: | AGAGTGTCCGAGTCCAGAGG (PET) | (AAG)11 | KX387449 | 72 | 199–217 | 6 | 0.736 | 0.683 | 0.892 |
| | R: | CAGTACCTGGTGATGGGAGC | | | | | | | | |
| Lethrinus laticaudis | Lel011 F: | CTGTCGGAGGTAAAGTGCG (FAM) | (AGC)9 | KX387422 | 84 | 237-285 | 11 | 0.655 | 0.660 | 0.251 |
| | R: | CTCATGGTGTTGAGGATGGG | | | | | | | | |
| | Lel040 F: | TGGTTGCAGACAACTGCC (VIC) | (AGC)9 | KX387431 | 84 | 171–216 | 14 | 0.821 | 0.825 | 0.367 |
| | R: | CTTAAGAGCAGTGATCCAGGC | | | | | | | | |
| | Lel033 F: | AGTGCGACAAAGAAATGGC (NED) | (AGAT)16 | KX387428 | 84 | 167–143 | 20 | 0.893 | 0.926 | 0.438 |
| | R: | CATTTGTCAGTTATGAAACTTGGC | | | | | | | | |
| | **Lel012 F:** | **GCGAGGGTCTGCTACTATAGGG (PET)** | **(AAT)9** | **KX387423** | **76** | **246-334** | **22** | **0.711** | **0.846** | **0.005** |
| | R: | TGTAAAGTGTAAACCACGTCCC | | | | | | | | |
| | Lel013 F: | CCTGAACCTGGAGAACTCGG (FAM) | (ATC)12 | KX387424 | 82 | 242–287 | 10 | 0.744 | 0.829 | 0.150 |
| | R: | ACTGAGGGAGGAGATAAAGGG | | | | | | | | |
| | Lel041 F: | CTGCTGTTCTGGGTTGCC (VIC) | (AAT)19 | KX387432 | – | – | – | – | – | – |
| | R: | CAACAAGCTGTTGGTGTCCC | | | | | | | | |
| | Lel032 F: | AAATCTGCATTATGAAATTGGC (NED) | (AAAG)16 | KX387427 | 83 | 173–233 | 16 | 0.867 | 0.883 | 0.209 |
| | R: | CAGCTCCTTGAGTTTAGTCCC | | | | | | | | |
| | Lel028 F: | CAGTAGCTTTAATAGTTAGGCACCC (PET) | (AAAG)13 | KX387426 | 83 | 200–244 | 14 | 0.843 | 0.877 | 0.954 |
| | R: | GGCTGTCCAGAGTGAGGC | | | | | | | | |
| | Lel047 F: | AAAGAATGGGAAGAATGACCC (FAM) | (AGAT)11 | KX387434 | – | – | – | – | – | – |
| | R: | AAGCCAAGTGATTAAGAAACCC | | | | | | | | |
| | Lel027 F: | CACTAAGGGTCCATGTTGCC (FAM) | (AAT)22 | KX387425 | 79 | 196–238 | 15 | 0.911 | 0.914 | 0.480 |
| | R: | TCTGTAATGAATGATCAAACCG | | | | | | | | |

Taillebois et al. (2016), PeerJ, DOI 10.7717/peerj.2418

Taillebois et al. (2016), PeerJ, DOI 10.7717/peerj.2418

**Table 3** (*continued*)

| Species | Locus | Primer sequences (5′–3′) (fluorescent label) | Repeat motif | GenBank accession no. | $n$ | Allele size range (bp) | #A | $H_O$ | $H_E$ | $P_{HW}$ |
|---|---|---|---|---|---|---|---|---|---|---|
| | *Lel044* | F: TTCTACTTGACCCTGGTAGGC (VIC) | (ATCC)11 | KX387433 | 83 | 151–199 | 11 | 0.759 | 0.816 | 0.268 |
| | | R: AATGTAATGCCATAAGCGGG | | | | | | | | |
| | *Lel036* | F: TCCAATTTACACCAAACTAGGC (NED) | (AAAG)15 | KX387429 | – | – | – | – | – | – |
| | | R: CCGGAATGATCTGCAGGC | | | | | | | | |
| | *Lel039* | F: CTTGTAGAGTGTCAACGAGGG (PET) | (AAT)11 | KX387430 | 80 | 196–214 | 9 | 0.700 | 0.764 | 0.747 |
| | | R: CATGATGCAATAACCATCCC | | | | | | | | |

**Notes.**

$n$ is the sample size, #A is the number of alleles at each loci, $H_E$ is the expected heterozygosity, $H_O$ is the observed heterozygosity and $P_{HW}$ is the $p$-value of the exact tests of conformance of genotypic proportions to Hardy-Weinberg equilibrium expectations.

**Table 4  Pairwise FST estimates for *Lutjanus johnii, Protonibea diacanthus,* and *Lethrinus laticaudis* for three locations.** Estimates are based on 10 microsatellite data from 68 individuals of *L. johnii*, 11 microsatellite data from 72 individuals of *P. diacanthus* and 10 microsatellite data from 84 individuals of *L. laticaudis* among the three sampling locations Camden Sound (CS), Wadeye (Wa) and Vanderlin Islands (VI). The comparisons that differed significantly from zero ($p < 0.05$) are shaded in grey.

| | *Lutjanus johnii* | | *Protonibea diacanthus* | | *Lethrinus laticaudis* | |
|---|---|---|---|---|---|---|
| | $F_{ST}$ | *p*-value | $F_{ST}$ | *p*-value | $F_{ST}$ | *p*-value |
| Wa-CS | 0.001 | 0.308 | 0.009 | 0.090 | **0.017** | **0.000** |
| Wa-VI | **0.007** | **0.043** | 0.007 | 0.069 | 0.007 | 0.069 |
| CS-VI | 0.006 | 0.148 | **0.014** | **0.013** | **0.009** | **0.017** |

and overall deviations from Hardy–Weinberg equilibrium (HWE) were detected at a single locus *Lel012* (*p*-value = 0.005). Heterozygosity was high and with an overall mean higher than for the two other species ($0.834 \pm 0.078$).

A pattern of genetic differentiation with low but significant population-pairwise $F_{ST}$ (range 0.007–0.017) was observed in the three species (Table 4). These levels of differentiation are similar in magnitude to those reported for other marine fish species with potentially high gene flow (*O'Reilly et al., 2004*). The three species also presented different patterns of structure between the three locations. *L. laticaudis* presented pairwise differentiation between Western Australia location (CS) and the two other Northern Territory locations (VI and Wa). Similarly, *P. diacanthus* presented a structure between the two most distant populations of CS and VI whereas CS and Wa remained undifferentiated. By contrast, *L. johnii* did not present structure between Western Australia and Northern Territory. However, VI and Wa in the Northern Territory had a significant pairwise $F_{ST}$. Pairwise $F_{ST}$ values were low in all comparisons for the three species (range 0.001–0.017) as commonly reported in marine species and, 4 out of 9 pairwise comparisons were significant meaning the microsatellites we developed can accurately detect population differentiation across Northern Australia and may be used for genetic structure studies and stock identification.

## CONCLUSION

In conclusion, we applied the direct sequencing of a genomic library approach to develop microsatellite loci and it resulted in a significant reduction in laboratory effort compared to traditional protocols for microsatellite discovery (cloning and Sanger sequencing). Merged paired-end reads from the MiSeq platform demonstrated higher quality of reads than the IonTorrent single reads. From the 1–1.5 million raw reads, we selected a reduced number of loci to test (48) and successfully amplified satisfactory polymorphic loci for 10 to 13 of them depending on the species. However, the NGS data revealed the potential for hundreds to thousands of potentially amplifiable microsatellites to be discovered. The microsatellites characterized in this work will be available to explore the population genetics and stock structure of these highly valuable species.

## ACKNOWLEDGEMENTS

The authors thank staff from the Western Australian Department of Fisheries and the Northern Territory Department of Primary Industries as well as Indigenous rangers for sample collection. We extend our thanks to Samuel Williams at the Molecular Fisheries laboratory and Sean Corley at the Animal Genetics Laboratory for their assistance with laboratory work.

### Funding

This research was supported by the Fisheries Research and Development Corporation (Project 2013/017); LT received additional funding support from the North Australia Marine Research Alliance (NAMRA—AIMS/ANU/CDU/NT Government) Postdoctoral Fellowship. The funders had no role in study design, data collection and analysis, decision to publish, or preparation of the manuscript.

### Grant Disclosures

The following grant information was disclosed by the authors:
Fisheries Research and Development Corporation: 2013/017.
North Australia Marine Research Alliance.

### Competing Interests

The authors declare there are no competing interests. David J. Welch is an employee of C2O Fisheries, Cairns, Queensland, Australia.

### Author Contributions

- Laura Taillebois conceived and designed the experiments, performed the experiments, analyzed the data, contributed reagents/materials/analysis tools, wrote the paper, prepared figures and/or tables, reviewed drafts of the paper.
- Christine Dudgeon performed the experiments, reviewed drafts of the paper.
- Safia Maher performed the experiments.
- David A. Crook, Diane P. Barton, Jonathan A. Taylor, David J. Welch and Richard J. Saunders reviewed drafts of the paper.
- Thor M. Saunders, Stephen J. Newman and Michael J. Travers contributed reagents/materials/analysis tools, reviewed drafts of the paper.
- Jennifer Ovenden conceived and designed the experiments, performed the experiments, analyzed the data, contributed reagents/materials/analysis tools, reviewed drafts of the paper.

### Animal Ethics

The following information was supplied relating to ethical approvals (i.e., approving body and any reference numbers):

Charles Darwin University Animal Ethics permit A13014.

## DNA Deposition

The following information was supplied regarding the deposition of DNA sequences:

GenBank

Lel011 KX387422
Lel012 KX387423
Lel013 KX387424
Lel027 KX387425
Lel028 KX387426
Lel032 KX387427
Lel033 KX387428
Lel036 KX387429
Lel039 KX387430
Lel040 KX387431
Lel041 KX387432
Lel044 KX387433
Lel047 KX387434
Luj068 KX387435
Luj076 KX387436
Luj094 KX387437
Luj091 KX387438
Luj082 KX387439
Luj051 KX387440
Luj027 KX387441
Luj114 KX387442
Luj072 KX387443
Luj090 KX387444
Prd012 KX387445
Prd018 KX387446
Prd020 KX387447
Prd023 KX387448
Prd024 KX387449
Prd036 KX387450
Prd042 KX387451
Prd044 KX387452
Prd045 KX387453
Prd046 KX387454
Prd049 KX387455.

## Data Availability

Laura Taillebois, Thor Saunders and Ovenden, Jennifer R. (2016): Next-generation sequencing data of black jewfish *Protonibea diacanthus*. The University of Queensland. Dataset. Available at http://espace.library.uq.edu.au/view/UQ:390570.

Laura Taillebois, Thor Saunders and Ovenden, Jennifer R. (2016): Next-generation sequencing data of grass emperor *Lethrinus laticaudis*. The University of Queensland. Dataset. Available at http://espace.library.uq.edu.au/view/UQ:390575.

Laura Taillebois, Thor Saunders and Ovenden, Jennifer R. (2016): Next-generation sequencing data of golden snapper *Lutjanus johnii*. The University of Queensand. Dataset. Available at http://espace.library.uq.edu.au/view/UQ:390574.

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
