# Peer review of "Characterization, development and multiplexing of microsatellite markers in three commercially exploited reef fish and their application for stock identification"

_PeerJ, doi:10.7717/peerj.2418_

## Round 0.1 · original submission · Major Revisions

I now have comments back from 2 referees on your submission, and while both agree that the work is of interest to the field and the microsatellites will be of value to future studies, they are split on whether or not your paper meets the criteria for acceptance at PeerJ. In particular, the second reviewer points out a number of issues in reaching your goal of comparing different NGS approaches in the paper. I find myself sharing their concerns about drawing conclusions and making strong statements with so many differences between the approaches. However this poses a problem, because this is essentially the only real question being addressed in the manuscript, and it is considered flawed by the referee. Thus, the most substantial issue that will need to be addressed in revision is the the biological question posed by the study.

The second referee points out that the manuscript is a very good primer note, but does not currently meet the PeerJ standards for publication, because it does not actually address a biological question. I find myself in agreement on both points, and I came away asking myself the same thing when reading the manuscript: what is the biological question being addressed by this submission that makes it more than a primer note? The manuscript does a good job of describing two species run through the IonTorrent and one through a MiSeq to obtain the microsatellites, the multiplex reactions used to score them, and the genetic diversity of these markers across each of the three species, but I see no specific biological question beyond this that convinces me this is more than a well done primer note for 3 species. The conclusion that hundreds of microsatellites can be isolated from genomic libraries sequenced by NGS is now common knowledge, and if the real question was to compare the Ion Torrent and Illumina platforms, you would have a more convincing case if you had run the same genomic DNA through each protocol and sequenced those libraries on the two NGS platforms to compare them directly.

Ultimately, I find myself in agreement with the second referee that if the authors wish to publish the work in PeerJ, they need to meet the specified standards outlined in the policies (see #9 at https://peerj.com/about/policies-and-procedures/#discipline-standards) that they address a biological question. With the information presented in the manuscript, it seems to me that it should be possible to expand the paper beyond a primer note and focus on some question of interest to the field, but I leave it to the authors as to which question, and how they wish to address that question in revision.

·

Basic reporting

The present study is a well-performed and well-written description of a microsatellite discovery and genotyping protocol development for three valuable fishery species, thus relevant for management.
The study is very straighforwad and I have no major comments, except for that a discussion section is missing. I would have welcome a explanation on why multiplexing did not perform well for one of the species (poor DNA quality?). Besides this and a few edits suggested in the attachment, I consider the article is worth publication and constitutes an interesting contribution to its field.

Experimental design

No major comments, but see attachment for minor suggestions

Validity of the findings

The results are robust

Reviewer 2 ·

Basic reporting

No Comments

Experimental design

The current version of the MS describes a suit of important genetic resources but does not address a relevant or meaningful research questions.

Validity of the findings

No coments

Additional comments

The MS by Taillebois and collaborators describes the development of 34 microsatellites for three species of commercially important reef fishes. The authors used two different next generation sequencing approaches to obtain between 1.5 and 2.8 million sequences per species from which microsatellite markers were characterized and tested in the laboratory.

Overall I found the MS to be well written and agreeable to read. The methods are described with sufficient detail and information to allow replication. However, my main concern with the current version of the MS is that it does not address a relevant and meaningful research question in Marine Biology. The current version of the MS is a good primer note (it describes in detail an important suit of microsatellite markers that I have no doubt will be useful for future studies), but it does not address any research questions in Marine Biology. Overall, I am afraid the MS does not meet the peerJ discipline specific standards (“Manuscripts that report the characterization of specific primers (such as microsatellites) should include substantial biological analyses.”) and should not be considered for publication in peerJ in its current version. If the authors are willing to introduce a significant biological question into the MS and address it, I encourage them to resubmit the MS to peerJ. If not, I would recommend submitting the MS to other journals such as Marine Biodiversity where primer notes are commonly accepted.

Specific comments:
Perhaps the only major comment I have is that, according to the last paragraph of the introduction (lines 75-80), one of the aims of this study was to compare and discuss the outcome in terms of number and quality of microsatellites obtained from different NGS strategies. I like the idea, but I am afraid the experimental design that was used is not appropriate to do this for various reasons. First, in order to rigorously compare two NGS strategies one should make sure every other possible source of variation is controlled for. In this case, not only you are using different species for different NGS sequencing approaches, but also different DNA extraction protocols were used for different NGS approaches. Under these circumstances, even if you find differences between NGS strategies, you cannot attribute the differences to this variable. Second, with an N=3, your experimental design is unbalanced and your sample size is extremely low to draw any robust conclusion regarding differences between NGS methods. Finally, your introduction states “we discuss different NGS approaches…” yet these are not actually addressed in the discussion. The results and discussion section is limited to describe the process of obtaining the final set of microsatellites per species and their attributes (Ho, He, HWE, LD, etc). Yet, there is nothing that addresses a biological problem, or a thorough discussion about the differences between NGS approaches in terms of output.

Lines 233-236. I find this result “the 13 loci for Le. Laticaudis did not amplify successfully as part of PCR multiplexes using the M13 labeling system” quite interesting. If one of the objectives of this MS is to discuss the differences between NGS approaches, then perhaps this is a result that should be discussed in detail. In particular because it suggests that the NGS approach has a strong impact on the success rate of marker multiplexing. However, as noted before, given that in the experimental design the DNA extraction method is also a variable, the authors should be careful about the conclusions drawn from these results.

---

## Round 0.2 · Minor Revisions

I have read through your revised manuscript, and in addition to the comments from the referees that you addressed in the revision, you have added a basic analysis of population structure to highlight the utility of the markers sufficient to meet the criterion of addressing a biological question for acceptance at PeerJ. I see no reason to delay the submission by returning it to the referees at this point given that you have addressed all of their comments, but there are some minor typographical errors or edits for clarification needed to the revised text before the paper can be published. I will email you the manuscript with tracked changes for my suggested edits so that you can decide if you wish to accept those suggestions or not and then upload a clean version (it's up to you, but it does not need to include tracked changes for me at this stage) to move forward into production.

---

## Round 0.3 · accepted · Accept

I am happy to now move your manuscript forward into production.